# New Phocoenamicin and Maklamicin Analogues from Cultures of Three Marine-Derived *Micromonospora* Strains

**DOI:** 10.3390/md21080443

**Published:** 2023-08-07

**Authors:** Maria Kokkini, Daniel Oves-Costales, Pilar Sánchez, Ángeles Melguizo, Thomas A. Mackenzie, Mercedes Pérez-Bonilla, Jesús Martín, Arianna Giusti, Peter de Witte, Francisca Vicente, Olga Genilloud, Fernando Reyes

**Affiliations:** 1Fundación MEDINA, Centro de Excelencia en Investigación de Medicamentos Innovadores en Andalucía, Parque Tecnológico Ciencias de la Salud, Avda. del Conocimiento 34, Armilla, 18016 Granada, Spain; daniel.oves@medinaandalucia.es (D.O.-C.); pilar.sanchez@medinaandalucia.es (P.S.); angelesmeguizomunoz1@gmail.com (Á.M.); thomas.mackenzie@medinaandalucia.es (T.A.M.); mercedes.perez@medinaandalucia.es (M.P.-B.); jesus.martin@medinaandalucia.es (J.M.); francisca.vicente@medinaandalucia.es (F.V.); olga.genilloud@medinaandalucia.es (O.G.); 2Laboratory for Molecular Biodiscovery, Department of Pharmaceutical and Pharmacological Sciences, University of Leuven, O & N II Herestraat 49-box 824, 3000 Leuven, Belgium; arianna.giusti0@gmail.com (A.G.); peter.dewitte@kuleuven.be (P.d.W.)

**Keywords:** antimicrobial resistance, drug discovery, natural products, marine actinomycetes, polyketides, spirotetronates, phocoenamicins

## Abstract

Antimicrobial resistance can be considered a hidden global pandemic and research must be reinforced for the discovery of new antibiotics. The spirotetronate class of polyketides, with more than 100 bioactive compounds described to date, has recently grown with the discovery of phocoenamicins, compounds displaying different antibiotic activities. Three marine *Micromonospora* strains (CA-214671, CA-214658 and CA-218877), identified as phocoenamicins producers, were chosen to scale up their production and LC/HRMS analyses proved that EtOAc extracts from their culture broths produce several structurally related compounds not disclosed before. Herein, we report the production, isolation and structural elucidation of two new phocoenamicins, phocoenamicins D and E (**1**–**2**), along with the known phocoenamicin, phocoenamicins B and C (**3**–**5**), as well as maklamicin (**7**) and maklamicin B (**6**), the latter being reported for the first time as a natural product. All the isolated compounds were tested against various human pathogens and revealed diverse strong to negligible activity against methicillin-resistant *Staphylococcus aureus*, *Mycobacterium tuberculosis* H37Ra, *Enterococcus faecium* and *Enterococcus faecalis*. Their cell viability was also evaluated against the human liver adenocarcinoma cell line (Hep G2), demonstrating weak or no cytotoxicity. Lastly, the safety of the major compounds obtained, phocoenamicin (**3**), phocoenamicin B (**4**) and maklamicin (**7**), was tested against zebrafish eleuthero embryos and all of them displayed no toxicity up to a concentration of 25 μM.

## 1. Introduction

Over the last three years, the COVID-19 pandemic has made public health a common subject of interest and concern worldwide and we have been reminded that the fight between humans and pathogens is always there. Antimicrobial resistance (AMR), one of the major global challenges in this fight, has also been affected by the pandemic and vice versa [1]. It is clear that some of the implications faced during the pandemic are directly related to AMR, as the role of bacterial co-infections in patients with severe COVID-19 has been crucial for their survival [2] and some of these infections are caused by nosocomial drug-resistant microorganisms, like methicillin-resistant *Staphylococcus aureus* and *Klebsiella pneumoniae* [3]. Therefore, the AMR landscape is likely to be different in the future and research has to be directed to determine and analyze these changes in the post-pandemic era and illustrate that AMR could be a hidden global pandemic [3].

Natural products (NPs) have played an important role in the fight against AMR. They offer exceptional chemical diversity and structural three-dimensional complexity which, having evolved over time [4], confers on them highly selective biological activities based on the hypothesis that all natural products have some receptor-binding function [5]. They are an unlimited source of new bioactive compounds, often inspiring the design and synthesis of analogues that may have improved properties [6].

Spirotetronates are a class of polyketide natural products produced by Actinomycetes. The first spirotetronate discovered back in 1969 was chlorothricin, produced by a *Streptomyces* strain isolated from a soil sample in Argentina [7]. Since then, more than 100 compounds of this structural class have been discovered. Structurally, the spirotetronates are characterized by a cyclohexene ring spiro-linked to a tetronic acid moiety and embedded in a macrocycle. The existence or not of a decalin unit groups them into class I (without) and class II (with the decalin unit). Further classification groups are based on the number of carbons of the macrocycle, in small (C_11_), medium (C_13_) and large spirotetronates (≥C_15_). Lastly, they often bear various oligosaccharide chains attached to the decalin and/or the macrocycle and different side chains that enrich their structural variety [7,8]. An exception to this classification is the quartromicins, unusual spirotetronate polyketides containing four spirotetronate subunits within one molecule [9].

Regarding their bioactivity, they have demonstrated a wide variety of biological activities, mostly antibiotic against Gram-positive bacteria and antitumor, but also antiviral, antiulcer and anti-inflammatory and found to be active against CNS diseases [7,8]. Along with their potent bioactivities, certain spirotetronates have been considered tools to elucidate a biological effect. For example, abyssomicin C was found to be the first natural product to block pABA biosynthesis. However, apart from some exceptions, there is no clear understanding of the role of the spirotetronate motif, the effect of the macrocyclic size, the oligosaccharide and side chains and the decalin system in the biological activities of the spirotetronates [10].

As part of the PharmaSea and MarPipe EU funded projects, phocoenamicins B and C, together with the known spirotetronate polyketide phocoenamicin [11], were isolated from cultures of *Micromonospora* sp. strain CA-214671 and demonstrated antimicrobial activity against methicillin-resistant *Staphylococcus aureus* (MRSA) and *Mycobacterium tuberculosis* H37Ra [12]. More recently, a survey of phocoenamicin producers from Fundación MEDINA’s strain collection identified 27 *Micromonospora* strains isolated from both terrestrial and marine environments. This survey eliminated previous statements about phocoenamicins production being unique to marine environments but in the meantime confirmed marine-derived strains as the best producers of the compounds in terms of abundance and variety of molecules [13]. Based on these results, the three marine-derived strains identified, namely *Micromonospora* sp., CA-214671, CA-214658 and CA-218877, were chosen to scale up their production and LC/HRMS analysis detected that their EtOAc extracts contain several previously undisclosed structurally related compounds. Herein, we report the isolation and structural elucidation of two new additional phocoenamicins, phocoenamicins D and E (**1**–**2**), along with the known phocoenamicin (**3**) and phocoenamicins B and C (**4**–**5**), maklamicin B (**6**) and maklamicin (**7**). Furthermore, the bioactivity against various human bacterial pathogens and inhibition of the cell viability by all the compounds isolated, as well as the safety against zebrafish eleuthero embryos of the main compounds was evaluated.

## 2. Results

### 2.1. Isolation and Taxonomy of the Producing Microorganisms

The producing strains CA-214671 and CA-214658 were isolated from marine cave sediments and CA-218877 from a marine invertebrate in Gran Canaria, Spain. Taxonomic identification was achieved by the sequencing of the 16S ribosomal gene with the universal primers fD1 and rP2. A BLASTN search in the EzBioCloud database of the PCR-amplified 16S rRNA nucleotide sequences strongly indicated that all three strains belong to the *Micromonospora* genus. The strains CA-214658 and CA-218877 are closely related to *Micromonospora endophytica* (99.63 and 100% similarity, respectively), and the strain CA-214671 to *M. chaiyaphumensis* (99.84% similarity) (Table 1).

### 2.2. Fermentation of the Producing Microorganisms, Extraction and Isolation of the Compounds

The three marine-derived strains were fermented for 14 days at 28 °C. Strain CA-214671 was fermented in 5 L of FR23 culture medium, while strains CA-214658 and CA-218877 in 3 L of RAM2-P V2 medium. The mycelium of the fermentation broths was separated from the supernatant by centrifugation, followed by filtration. Both phases were extracted with ethyl acetate, the mycelium using a magnetic stirrer (190 rpm, 2 h) and the supernatant by liquid–liquid extraction in a separatory funnel. The combined organic phases were then evaporated to provide the crude extracts, which were fractionated through reversed-phase C18 medium-pressure chromatography. LC-UV-MS analysis of the fractions revealed the presence of various known spirotetronates and minor amounts of related compounds that suggested their novelty as natural products, reaffirming the results of our previous work [13].

Hence, further chromatographic separation by several steps of preparative and semi-preparative reversed-phased HPLC on a phenyl and pentafluorophenyl column, led to the isolation of seven compounds (**1**–**7**), including the two new phocoenamicins D and E (**1**–**2**), together with the known phocoenamicin, phocoenamicins B and C (**3**–**5**), maklamicin B (**6**) and maklamicin (**7**), also belonging to the spirotetronate class of compounds (Figure 1).

### 2.3. Structural Elucidation

Compound **1** was isolated as a white amorphous solid. A molecular formula of C_56_H_75_ClO_20_ was deduced from the (+)-ESI-TOF analysis that displayed an adduct ion at *m*/*z* 1120.4884 [M + NH_4_]^+^, accounting for 19 degrees of unsaturation. IR absorptions at 3376, 1776, 1676 and 1445 cm^−1^ were indicative of the presence of hydroxy, carbonyl and olefinic groups. The molecular formula together with the characteristic UV absorption pattern with maxima at 230, 290 and 320 nm strongly suggested a phocoenamicin-related structure for the compound.

Its ^13^C NMR spectrum revealed the presence of three esters at δ_C_ 169.2, δ_C_ 175.6, 165.6 and one ketone at 215.4, together with five methylene sp^3^ carbons, ten methyl groups and numerous methine and quaternary sp^2^ and sp^3^ carbons (Table 2). The ^1^H NMR data, in combination with the HSQC spectrum, revealed that **1** contains a substituted benzoic acid moiety and two deoxysugar units, characteristic for this family of compounds (Table 2).

Compound **1** exhibited almost identical ^13^C and ^1^H NMR spectroscopic data than those described for phocoenamicin C (**5**) [12], the major difference between the ^13^C NMR spectra of the two compounds being the presence of an oxygenated methylene carbon at δ_C_ 65.1 (C-30) in **1** (Table 2), instead of the methyl group at δ_C_ 22.4 observed in phocoenamicin C (**5**). This difference was confirmed in the HSQC and ^1^H NMR spectra with the presence of two oxygenated methylene hydrogens at δ_H_ 4.03 and 4.14 in **1** that correlated in the HSQC spectrum with the carbon at δ_C_ 65.1 (Table 2) instead of the methyl group at δ_H_ 1.73 ppm in phocoenamicin C (**5**). Two-dimensional HMBC correlations between H-19 and C-30, and between both H_2_-30 and C-19, C-20 and C-21 confirmed that a hydroxymethyl group is located at C-20 (Figure 2). The same hydroxymethyl functionality at C-20 is observed in phocoenamicin B (**4**) [12], indicating a functional group conserved within the family. The difference was in agreement with the proposed molecular formula for compound **1**, having one oxygen more than **5**.

COSY and HMBC correlations (Figure 2) confirmed phocoenamicin D (**1**) to have the same carbon skeleton as **5** and the linkages between the oxygenated C-9 of the *trans*-decalin substructure and C-1′ of one of the deoxyglucose units, between C-3′ of this deoxyglucose and C-1″ of a second deoxyglucose unit and finally between C-4″ of the latter and the carbonyl group (C-7‴) of the 3-chloro-6-hydroxy-2-methylbenzoate moiety. Additionally, NOESY correlations confirmed the compound to have the same configuration in all its chiral centers as the previously proposed for other members of the phocoenamicin family (Figure 3). Particularly, correlations between H-5, H-7β and H-9 and between H-6, H-8 and H-10 suggested a trans ring fusion of the decalin unit and further correlations between H-7α and H-26 and H-27 placed the methyl groups C-26 and C-27 at equatorial orientation. Additional NOEs between H-10 and H-25 placed the C-25 methyl group in the bottom face of the molecule. A series of NOESY correlations (H-13/H-15, H-14/H-16, H-15/H-17α, H-17β/H-19) determined a zigzag conformation of the C-13 to C-18 chain (Figure 3). Furthermore, the ^3^*J*_HH_ coupling constants between H-11 and H-12 (9.3 Hz) and between H-15 and H-16 (14.6 Hz) assigned the olefins as Z and E, respectively. NOE correlations from H-15 to H-28 placed the methyl group C-28 on the bottom face and from H-16 to H-29 placed C-29 on the top face of the molecule. The chair conformation of the cyclohexene ring was determined by NOEs between H-29, H-21 and H-22β that placed them on the same side of the ring (Figure 3). The configuration of the diol side chain and tetronic acid could not be assigned due to extensive signal overlapping and was hypothesized to be identical with the rest of phocoenamicins, setting a configuration S*/R* at C-32 and C-33 carbons and an S* configuration for the tetronic acid stereogenic center, respectively [11,12]. Finally, NOESY correlations in the sugar units (Figure 3) established the axial orientation of all protons, confirming two units of β-6-deoxyglucopyranoside, whose absolute configuration is proposed as D in alignment with phocoenamicin (**3**) and other members of the family. [11,12]. The name phocoenamicin D was proposed for compound **1**.

Compound **2** was obtained as a white amorphous solid, whose (+)-ESI-TOF analysis identified a protonated adduct [M + H]^+^ at *m*/*z* 1015.4462, thus giving a molecular formula of C53H71ClO17, which indicated 18 degrees of unsaturation. IR absorptions at 3388, 1743, 1677 and 1445 cm^−1^ suggested again the presence of hydroxy, carbonyl and olefinic functionalities.

This molecular formula had three carbons, four hydrogens and three oxygen atoms less than that of **1**. The NMR data of **2** were very similar to those of **1** (Table 2). The major differences were observed in the side chain attached to C-21, where the signals of the quaternary oxygenated carbon C-33, the ketone C-34 and the methyl groups C-35 and C-36 in **1** were replaced by a doublet methyl signal (δ_C_ 1.18, δ_C_ 24.6) in **2**. Comparing with the NMR data obtained for **1**, the presence of a 2-hydroxy-1-propyl group attached to C-21 was further supported by the differences in the chemical shifts of carbons C-31 (δ_C_ 33.8 to 42.4), C-32 (δ_C_ 74.0 to 66.0) and the presence of a methyl group instead of a quaternary sp^3^ carbon at C-33 (δ_C_ 83.4 to 24.6), as well as the COSY correlations from H-31 to H-32 and from H-32 to H-33 and the HMBC correlations from H-31 to C-22 and C-33. Furthermore, a ketone carbonyl signal was observed at C-3 (δ_C_ 201.5) in **2**, instead of the ester carbonyl carbon at δ_C_ 175.6 present in **1,** confirmed through an intense HMBC correlation to methyl C-25.

The rest of the NMR data were similar to those of **1**, confirming the presence of a spirotetronate with an eleven-membered macrocycle core, a *trans*-decalin unit and a disaccharide connected to a substituted benzoic acid moiety (Figure 1). Indeed, apart from a different side chain at C-21, the structure of the molecule is identical to that of phocoenamicin B (**4**), having a ketone group at C-3 and a hydroxymethyl group at C-20. The hydroxyethyl side chain at C-21 was previously reported as a structural element of maklamicin, a closely related spirotetronate [14].

Similar to compound **1**, the relative configuration of the chiral centers of **2** was determined by a combination of NOESY correlations (see Appendix A) and ^3^*J*_HH_ coupling constants and was found to be identical to that of **1**. Regarding the hydroxyethyl side chain at C-31, and similarly to what is reported for maklamicin, NOESY correlations between H-22α/H-32 and H-31α/H-30 and a large ^3^*J*_HH_ coupling constant between H-31α and H-32 (9.6 Hz) and between H-31β/H-21 (as indicated by an intense COSY crosspeak), set an anti-relationship between H-31α and H-32, as well as between H-31β and the 32-OH group and assigned an *R** configuration at C-32, the same configuration described for phocoenamicins and maklamicin [11,14]. Similar chemical shift values around this side chain in the NMR spectra of compounds **2**, **6** and **7** confirmed this stereochemical assignment. The name phocoenamicin E was proposed for the compound.

Compounds **6** and **7** were isolated as white amorphous solids. The (+)-ESI-TOF spectrum of **6** showed a protonated adduct at *m*/*z* 509.3275 [M + H]^+^, accounting for a molecular formula of C_32_H_44_O_5_, while the (+)-ESI-TOF spectrum of **7** showed ions at *m*/*z* 525.321 [M + H]^+^ and 542.3486 [M + NH_4_]^+^ assigning a molecular formula of C_32_H_44_O_6_, thus revealing a difference of one oxygen atom between them.

Their UV, IR and NMR spectroscopic data were almost identical and shared common features. The UV absorption pattern (λ_max_ 230 and 310 nm), the IR spectrum of both (broad absorption bands for multiple hydroxy, carbonyl and olefinic groups), along with their molecular formulae suggested that these compounds were related to the spirotetronate maklamicin, with compound **7** sharing the same molecular formula and spectroscopic data, confirming its identity as maklamicin, and **6** lacking one oxygen atom.

Analysis of the ^1^H NMR and ^13^C NMR spectra of **6** (Table 3) identified the presence of 32 carbons, corresponding to three oxygen-bearing quaternary sp^2^ carbons (δ_C_ 205.0, 202.4 and 177.5), three quaternary sp^3^ carbons, seven sp^2^ carbons, six sp^3^ methylenes, seven sp^3^ methines and six methyl groups.

Interpretation of 1D and 2D NMR spectroscopic data gave the full planar structure of **6** (Figure 1). HSQC and COSY spectra constructed the different fragments of the compound and HMBC cross-peaks linked the fragments together, confirming a maklamicin-related compound (Figure 3).

Firstly, a decalin unit was established through COSY (H-5/H-6, H-7/H-8/H-9/H-10 and H/11/H-12/H-13) and plenty of HMBC correlations (C-4/H-12/H-13/H-25, C-5/H-9/H-11/H-25, C-7/H-5/H-26 and C-9/H-8/H-10/H-11/H-26), containing two methyl groups at C-4 and C-8 and a double bond at C-11/C-12 (Figure 4). The chain was extended through sequential COSY cross-peaks from H-13 to H-17 with a methyl group at C-14 and a double bond at C-15/C-16. Then a cyclohexene ring was determined through HMBC correlations (C-18/H-17/H-22 and C-23/H-17/H-19/H-21), containing two methyl groups at C-18 and C-20, as well as a side chain at C-21 (Figure 3). The ^13^C NMR spectrum showed characteristic of tetronic acid carbons resonances, connected to a ketone group at C-3, which completed the structure. A HMBC correlation from H-17 to C-23 confirmed the presence of a spirocenter in the tetronic acid (C-23). Finally, the structure of the side chain was elucidated as a 2-hydroxypropyl unit attached to C-21, similar to that found in phocoenamicin E (**2**) and maklamicin (**7**). Sequential COSY cross peaks (H-21/H-30/H-31/H-32), as well as HMBC correlations from H-30 to C-22 and C-32 and from H-31 to C-21, confirmed this proposal (Figure 3).

The major difference between compounds **6** and **7**, based on their NMR spectra, was the presence of a methyl group at δ_C_ 22.3 and δ_H_ 1.75 in **6**, instead of a hydroxymethyl group at δ_C_ 65.1 and δ_H_ 4.16, 4.01 observed in **7**. HMBC cross-peaks between the carbon of this methyl group and H-19, H-22 and H-30 and between its methyl protons and C-19, confirmed that the methyl group was located at C-20. Furthermore, the chemical shifts of C-20 and C-21 of **6** were slightly shifted to 135.0 and 34.5 ppm compared to 138.9 and 30.5 ppm in **7**, respectively (Table 3).

Finally, the relative configuration of **6** was elucidated by ROESY experiments (see Appendix A) and ^3^*J*_HH_ coupling constants and determined to be the same as that of maklamicin. The latter was supported by the almost coincident NMR chemical shift values determined for most of the hydrogens and carbons of the molecule, except those around C-29, where the major difference between both compounds was found. Particularly, in the case of the side chain at C-21 and in line with previous findings in maklamicin, NOESY correlations observed between H-22α/H-31 and H-30β/H-29 (Figure 5) together with the large ^3^*J*_HH_ coupling constants between H-21/H30b) and H-30a/H-31 set an anti-relationship between H-30α and H-31, as well as between H-30β and the 31-OH group and assigned an *R** configuration at C-31, the same configuration found in **2** and maklamicin [14]. Finally, the configuration for the tetronic acid chiral center was tentatively assigned as *S** in agreement with that of the previously reported spirotetronate antibiotics [10].

Compound **6** was previously reported as a result of gene deletion (*makC2*) in a genetically engineered strain and an intermediate in the biosynthetic pathway of maklamicin [14]. Herein, it is reported for the first time as a natural product, therefore proposing the name maklamicin B.

### 2.4. Antimicrobial and Cytotoxic Activity

The seven compounds isolated were tested in parallel against a panel of human pathogens, including methicillin-resistant *S. aureus* (MRSA) MB5393, *M. tuberculosis* H37Ra (ATCC25177), *E. faecium* (VANS 144754), *E. faecalis* (VANS 144492) and *N. gonorrhoeae* (ATCC49226). Overall, the eight spirotetronates exhibited strong to negligible activity, depending on the compound and the pathogenic bacterium, as shown in Table 4, expressed as minimum inhibitory concentration (MIC) values. Additionally, their cytotoxicity was evaluated in vitro against the human liver adenocarcinoma cell line (Hep G2) and demonstrated no cytotoxicity in most cases, except for the compounds **2**, **4** and **7** that presented moderate activity (Table 4).

The compounds tested had, in many cases, small structural differences when compared to each other, therefore the antibacterial assays revealed structural features that may influence the bioactivity and provided some structure–activity relationship conclusions.

Maklamicin (**7**) and maklamicin B (**6**) strongly inhibited the growth of MRSA with MIC values of <0.5 and 2 μM, respectively and comparable to vancomycin (1.4 μM), followed by phocoenamicin (**3**) and phocoenamicin B (**4**) (3.7 and 7.4 μM, respectively). The carboxylic ester instead of a ketone group in the macrocycle core (C-3) (Figure 1) observed in phocoenamicin C (**5**) and D (**1**) seemed to weaken the activity against MRSA, as previously reported [12].

Interestingly, the most active compound against *M. tuberculosis* H37Ra was maklamicin B (**6**), followed by phocoenamicin (**3**) showed significant activities with MIC values of 2 and 7.5 μM, respectively, while moderate to weak activity was observed in phocoenamicin B (**4**) and maklamicin (**7**) (14.7 and 61 μM, respectively), indicating that the lack of hydroxylation at C-29 increased the activity against this pathogen. The positive control streptomycin exhibited less activity (2.8–5.4 μM) than maklamicin B (**6**). To the best of our knowledge, the activity of maklamicin (**7**) and maklamicin B (**6**) against *M. tuberculosis* H37Ra has not been studied before.

On the contrary, the bioactivity assay against *E. faecalis* showed that the presence of hydroxylation at C-29 enhances the bioactivity of maklamicin (**7**) (1.9 μM), comparable to vancomycin (1.4 μM), in comparison to maklamicin B (**6**) (62.9 μM). The bioactivity of phocoenamicins against this pathogen has not been studied before and showed that phocoenamicin (**3**) exhibited significant activity with a MIC value of 7.5 μM, while the rest of the phocoenamicins demonstrated weak or no activity at the highest concentration tested. Maklamicin B (**6**) and maklamicin (**7**) also demonstrated strong activity against *E. faecium* with MIC values of 1 μM for both compounds, not reported before and stronger than vancomycin (1.4 μM), followed by phocoenamicin (**3**) (3.7–7.5 μM). Finally, none of the compounds inhibited the growth of the Gram-negative pathogen *N. gonorrhoeae* at the highest concentration tested.

Furthermore, the MTT toxicity assays demonstrated that only compounds **2**, **4** and **7** presented moderate cytotoxicity against the Hep G2 cell line, with 50% inhibitory concentration (IC_50_) values of 21, 17.6 and 40.2 μM, respectively, while the remaining compounds displayed no cytotoxicity at the highest concentration tested, making the most active of them potential candidates for further research and development.

### 2.5. Zebrafish Eleuthero Embryos Toxicity Assay

Zebrafish (*Danio rerio*) is a small tropical fish that lives in the freshwater rivers and lkes of South Asia and is characterized by the dark blue stripes covering its adult body [15,16,17]. Both adults and embryos are popular as laboratory models in biological research but embryos and larvae are mostly used to evaluate the toxicity of compounds due to their high sensitivity and are particularly useful in the early stages of preclinical studies [18,19]. Their usage is in compliance with the 3R guiding principles that highlight the importance of replacing animals like rodents with lower-order animals. Moreover, they are legally not considered an animal during the first five days post fertilization (dpf). They can be considered as a step between in vitro cell-based models and in vivo mammalian experiments [19].

Their small size (1–5 mm) allows the performance of miniature in vivo experiments in multi-well plates and high throughput screening. As a result, the amounts of compound required for these experiments are very low, making zebrafish attractive to natural product drug discovery, where the availability of compounds is usually limited [20]. Moreover, in these early life stages, their body is transparent, enabling direct observation using a simple stereo microscope [21].

The toxicity of the three major compounds isolated, phocoenamicin (**3**), phocoenamicin B (**4**) and maklamicin (**7**) was evaluated using zebrafish eleuthero embryos and the immersion method was used, as suggested for lipophilic compounds that have logD > 1 and are better absorbed by the zebrafish larvae [22,23].

The compounds were dissolved in dimethyl sulfoxide (DMSO) and then diluted in the fish medium (Danieau) in which the 3 dpf zebrafish larvae were swimming for 48 h. The concentrations tested were 1.6, 3.13, 6.25, 12.5 and 25 μM for each of the three compounds. Vehicle-treated Control eleuthero embryos (VHC) were also treated with the same concentration of DMSO (0.5%) in fish medium to compare the possible effects of the compounds.

After the 48 h exposure, corresponding to 5 dpf larvae, the possible mortality, touch-stimulation response and developmental abnormalities were examined using a dissecting microscope coupled with a digital color camera and pictures were captured. Then, the score for the sub-lethal toxicity and lethality was determined for each fish and condition and the mean score was calculated. The assays were independently replicated three times, using 10 embryos for each condition and compound in each assay, and thus the total number of zebrafish embryos used was 480.

The results demonstrated that the mean score of toxicity was low for all three compounds (Figure 6). In a small number of eleuthero embryos, abnormalities were observed that included bad development, body curvature and swim bladder defects. However, these abnormalities were not dose-dependent and did not differ significantly from the control group (*p* < 0.05). Therefore, the three spirotetronates were considered not toxic in a wide range of doses (1.6–25 μM), indicating the developing embryo’s ability to process and eliminate them. In Figure 7, representative zebrafish larvae with no developmental abnormalities observed are shown for each compound after the exposure at the highest concentration of 25 μM, as well as for the control group (VHC). To the best of our knowledge, this is the first assessment of the toxicity of spirotetronates using zebrafish eleuthero embryos.

Finally, the highest concentration tested (25 μM) was correlated to the results of the antimicrobial and cytotoxicity assays (Section 2.3). In all cases, these concentrations were higher than the concentrations where the three compounds were found to display moderate or strong antibacterial activities. Furthermore, for maklamicin (**7**) and phocoenamicin B (**4**) that demonstrated moderate cytotoxicity, the cytotoxic concentrations are comparable (**4**) or higher (**7**) than those used in the zebrafish embryos toxicity assay, suggesting no broad-spectrum toxicity at the highest concentration tested.

## 3. Discussion

Three marine-derived strains, isolated from marine cave sediments and a marine invertebrate in Gran Canaria, Spain and whose PCR-amplified 16S rRNA nucleotide sequences strongly indicated that they belong to the *Micromonospora* genus, were cultivated and led to the isolation of seven compounds (**1**–**7**), including two new phocoenamicins (**1**–**2**), together with the known phocoenamicin, phocoenamicins B and C (**3**–**5**). Furthermore, two additional spirotetronates, maklamicin (**7**) and maklamicin B (**6**) were isolated. Maklamicin B (**6**) was previously reported as a result of gene deletion (*makC2*) in a genetically engineered strain [23] and is herein reported as a natural product for the first time.

The two families of compounds share many structural features as they contain a tetronic acid spiro-linked to a cyclohexene ring, embedded in an eleven-carbon macrocycle and connected to a *trans*-decalin moiety. Their major differences are the presence (phocoenamicins) or not (maklamicins) of a disaccharide linked to a chlorinated phenol, as well as the characteristic diol side chain unique in phocoenamicins. Interestingly, phocoenamicin E (**2**) bore instead the same side chain as maklamicin (**7**) at C-21, not reported before in the phocoenamicin family, which suggested the existence of a common biosynthetic pathway in the production of the two families of compounds.

Common structural variations shared by the isolated compounds include a hydroxymethylene group (phocoenamicin B, D, E and maklamicin) or a methyl group (phocoenamicin, phocoenamicin C, and maklamicin B) located at C-20, a diol side chain (phocoenamicin, phocoenamicins B, C and D) or a 2-hydroxy-1-propyl group (phocoenamicin E and maklamicins) located at C-21, a ketone group (phocoenamicin, phocoenamicin B and E, maklamicins) or ester group (phocoenamicins C and D) located at C-3 and attached to the tetronic acid, indicating functional groups conserved within the two families. Among other structurally related spirotetronates, the hydroxymethylene or methyl group at C-20 can also be found in the nomimicin [24], lobophorin [25] and kijanimicin [26] families of compounds and the ester group at C-3 in chlorothricins [27] and PA-46101 A and B [28].

The bioactivity of the seven compounds isolated was evaluated against a panel of human pathogens, including methicillin-resistant S. aureus (MRSA), *M. tuberculosis* H37Ra, *E. faecium*, *E. faecalis* and *N. gonorrhoeae*. In similarity to other related spirotetronates, most of the compounds showed antibacterial activity against Gram-positive bacteria. Overall, the seven spirotetronates exhibited strong to negligible activities, depending on the compound and the pathogen. As mentioned above, the compounds had small structural differences when compared to each other, therefore the antibacterial assays revealed some structure–activity relationships.

The growth of MRSA was strongly inhibited by maklamicin (**7**) and maklamicin B (**6**), followed by phocoenamicin (**3**) and phocoenamicin B (**4**). The ester instead of a ketone group located at C-3, which was present in phocoenamicin C (**5**) and D (**1**) seemed to weaken the activity against MRSA, as previously reported [12]. Other structurally related families of spirotetronates that have demonstrated strong activities against MRSA are decatromicins [29], the compounds JK [30], as well as the recently discovered glenthmycins [31].

The most active compound against *M. tuberculosis* H37Ra was maklamicin B (**6**), followed by phocoenamicin (**3**), while moderate to weak in phocoenamicin B (**4**) and maklamicin (**7**), suggesting that the methyl instead of hydroxymethylene group in the side chain (C-20), strengthened the activity against the resistant pathogen. To the best of our knowledge, the activity of maklamicin (**7**) and maklamicin B (**6**) against *M. tuberculosis* H37Ra has not been studied before. Other related families of spirotetronate polyketides that have exhibited strong activities against *M. tuberculosis* are lobophorins [32], and the above-mentioned glenthmycins [31].

On the contrary, the presence of the hydroxymethylene group at C-20 may have enhanced the activity against *E. faecalis*, as shown by the results obtained with maklamicin (**7**) (1.9 μM) in comparison to maklamicin B (**6**) (62.9 μM). The bioactivity of phocoenamicins against this pathogen has not been studied before and showed that phocoenamicin (**3**) exhibited significant activity, while the rest of the phocoenamicins demonstrated weak or no activity at the highest concentration tested. The spirotetronate MM46115 [33] and two of the glenthmycins [31] have exhibited bioactivity against *E. faecalis* comparable to that of maklamicin (**7**). Furthermore, both **6** and **7** exhibited strong activity against *E. faecium* (1 μM), not reported before, followed by **3** (3.7–7.5 μM), expanding the panel of antimicrobial properties of these compounds. Overall, the lack of the glycosidic side chain in maklamicins (**6**) and (**7**) seems to increase the antimicrobial activity compared to that of phocoenamicins in some of the tested pathogens. Finally, none of the compounds inhibited the growth of the Gram-negative *N. gonorrhoeae* at the highest concentration tested.

Phocoenamicins C, D and E (**5**, **1**–**2**) demonstrated weak to negligible activity against the pathogenic bacteria tested. All of them had either the hydroxymethylene group in the side at C-20 (**2**) or the carboxylic ester in the macrocycle core (C-3) **(5**) or both of these structural features (**1**) that, as mentioned above, based on the structure–activity relationship findings, may affect the bioactivity against these pathogens. Additional antimicrobial assays should be performed studying the activity against other pathogens, based on the hypothesis that all-natural products have some receptor-binding function [5], as in the case of phocoenamicin (**3**) against *Clostridium difficile*, where the chlorosalicyclic ester seems to play an essential role [11].

Furthermore, the cytotoxicity of the compounds was evaluated in vitro against the human liver adenocarcinoma cell line (Hep G2) and demonstrated no cytotoxicity, except for the compounds **2**, **4** and **7** that presented moderate activity, making the most active of the compound’s potential candidates for further research and development.

Finally, zebrafish eleuthero embryos were used to evaluate the toxicity of the three major compounds, phocoenamicin (**3**), phocoenamicin B (**4**) and maklamicin (**7**). The three spirotetronates were considered not to be toxic in a wide range of doses (1.6–25 μM) and the highest concentration tested (25 μM) on zebrafish larvae was correlated to the results of the antimicrobial and cytotoxicity assays. In all cases, the testing concentrations were higher than the concentrations where the three compounds were found to display strong or moderate bioactivities against the pathogenic bacteria. Moreover, for maklamicin (**7**) and phocoenamicin B (**4**) that demonstrated moderate cytotoxicity, the cytotoxic concentrations were comparable to (**4**) or higher (**7**) than those used in the zebrafish embryos toxicity assay, suggesting no broad-spectrum toxicity in the highest concentration tested. To the best of our knowledge, this is the first toxicity assay using zebrafish eleuthero embryos on spirotetronates.

The new analogues isolated highlight the wide range of structural possibilities of the spirotetronates that could unveil new biological activities. The possible structure–activity relationships recorded here, along with others mentioned before within the spirotetronate class, can be used as pieces in the puzzle to get to the remarkable activities that these compounds can give.

## 4. Materials and Methods

### 4.1. General Experimental Procedures

LC-UV-MS analysis was performed on an Agilent 1100 (Agilent Technologies, Santa Clara, CA, USA) single quadrupole LC-MS system as previously described [34]. ESI-TOF and MS/MS spectra were acquired using a Bruker maXis QTOF (Bruker Daltonik GmbH, Bremen, Germany) mass spectrometer coupled to an Agilent 1200 LC (Agilent Technologies, Waldbronn, Germany). Medium-pressure liquid chromatography (MPLC) was performed on semiautomatic flash chromatography (CombiFlash Teledyne ISCO Rf400×) with a precast reversed-phase column. HPLC separations were performed on a Gilson GX-281 322H2 (Gilson Technologies, Middleton, WI, USA) using a semi-preparative reversed-phase column (XBridge prep Phenyl 5 μm, 10 × 150 mm) or a preparative (Kinetex^®^ 5 μm PFP 100 Å AXIA Packed LC Column, 250 × 21.20 mm) reversed-phase column. Solvents used for the extraction were of analytical grade and those used for the isolation were of the HPLC grade. In brief, 1D- and 2D-NMR spectra were recorded on a Bruker Avance III spectrometer (500 and 125 MHz for ^1^H and ^13^C NMR, respectively) equipped with a 1.7 mm TCI MicroCryoProbe^TM^ (Bruker Biospin, Fällanden, Switzerland). Chemical shifts were reported in ppm using the signals of the residual solvents as internal reference (δ_H_ 3.31 and δ_C_ 49.15 for CD_3_OD). Optical rotations were measured on a Jasco P-2000 polarimeter (JASCO Corporation, Tokyo, Japan). IR spectra were recorded with a JASCO FT/IR-4100 spectrometer (JASCO Corporation) equipped with a PIKE MIRacle^TM^ single reflection ATR accessory. Molecular models were generated using Chem&Bio Draw 12.0 (CambridgeSoft, PerkinElmer Informatics, Waltham, MA, USA).

### 4.2. Taxonomical Identification of the Producing Microorganisms

The taxonomic identification of the three strains was performed as previously described [13,35].

### 4.3. Fermentation of the Producing Microorganisms

A 5 L fermentation of the CA-214671 strain was generated as follows: a fresh seed culture of the strain was obtained inoculating a 25 × 150 mm tube containing 16 mL of ATCC-2-M medium (soluble starch 20 g/L, glucose 10 g/L, NZ Amine Type E 5 g/L, meat extract 3 g/L, peptone 5 g/L, yeast extract 5 g/L, sea salts 30 g/L, calcium carbonate 1 g/L, pH 7) with a freshly thawed inoculum stock of the strain. The tube was incubated for 7 days in an orbital shaker at 28 °C, 220 rpm and 70% relative humidity. The grown culture was then used to inoculate three flasks, each containing 50 mL of ATCC-2-M medium (5% *v*/*v*). The flasks were again incubated for seven days in an orbital shaker under the same conditions. The content of the three flasks was then mixed, and the mixture was used to inoculate 100 flasks, each containing 50 mL of FR23 fermentation medium (glucose 5 g/L, soluble starch from potato 30 g/L, cane molasses 20 g/L, cottonseed flour 20 g/L, sea salts 30 g/L, pH 7) (2.5% *v*/*v*). The flasks were incubated likewise for 14 days before harvesting.

For strains CA-214658 and CA-218877, a similar protocol without sea salts in the seed and fermentation media was employed. Fresh seeds of strains CA-214658 and CA-218877 in ATCC-2 medium (soluble starch 20 g/L, glucose 10 g/L, NZ Amine Type E 5 g/L, meat extract 3 g/L, peptone 5 g/L, yeast extract 5 g/L, calcium carbonate 1 g/L, pH 7) were used to inoculate (2.5% *v*/*v*) sixty flasks (3 L fermentation) of RAM2-P V2 (glucose 10 g/L, maltose 15 g/L, corn meal yellow 4 g/L, bacto yeast extract 5 g/L, proteose peptone 5 g/L, pH 7). Flasks were incubated at 28 °C, 220 rpm and 70% relative humidity for 14 days before harvesting.

### 4.4. Extraction and Isolation of the Compounds

The fermentation broth (5 L) from strain CA-214671 was extracted as follows: First, the separation of the mycelium and supernatant was achieved by centrifugation at 9000 rpm for 10 min, followed by filtration under vacuum. The supernatant was then subjected to liquid–liquid extraction with EtOAc in a separatory funnel in a ratio 1:1 and rotary evaporated until dry. The mycelium was extracted with 700 mL EtOAc in a magnetic stirrer (190 rpm, 2 h), filtered in a Büchner funnel and evaporated until dry. The extraction procedure was repeated in triplicate to obtain the final organic extract (6.48 g).

Reversed Phase C-18 silica gel was mixed with the final organic extract in a 1:2 ratio and loaded onto a C-18 column (ODS) (200 × 35 mm) that was eluted (MPLC) with a linear H_2_O-CH_3_CN gradient (10 mL/min; 5–100% CH_3_CN in 60 min; UV detection at 210 nm and 280 nm). (5% to 100% CH_3_CN in 35 min + 100% CH_3_CN in 25 min, 10 mL/min) to afford 65 fractions.

They were combined into seven groups according to their LC-UV-MS profiles and evaporated to dryness in a centrifugal evaporator: fractions A (127.8 mg), B (54.1 mg), C (39.4 mg), D (49.8 mg), E (64.9 mg), F (51.6 mg) and G (164 mg). Fractions containing the compounds of interest from this chromatography were further purified by preparative and semi-preparative reversed-phase HPLC.

Fraction A (127.8 mg) was chromatographed by preparative reversed-phase HPLC (Kinetex^®^ 5 μm PFP 100 Å AXIA Packed LC Column, 250 × 21.20 mm; 14 mL/min, UV detection at 210 and 280 nm) with a linear H_2_O-CH_3_CN with 0.1% trifluoroacetic acid gradient of 40–100% in 40 min yielding **1** (1.1 mg, t_R_ 12 min) and **4** (6.1 mg, t_R_ 18 min).

Fraction C (39.4 mg) was chromatographed by semi-preparative Reversed Phase HPLC (XBridge prep Phenyl 5 μm, 10 × 150 mm; 3.6 mL/min, UV detection at 210 and 280 nm) with an isocratic elution of 50% CH_3_CN/50% H_2_O with 0.1% trifluoroacetic acid over 40 min yielding **2** (1.0 mg, t_R_ 19 min).

The fermentation broth from strain CA-214658 was extracted with EtOAc as mentioned above for strain CA-214671 and MPLC afforded 39 fractions that were combined into five groups according to their LC-UV-MS profiles. Τhose containing the compounds of interest from this chromatography were further purified by preparative and semi-preparative reversed-phase HPLC.

Fraction A (285.9 mg) was chromatographed by semi-preparative Reversed Phase HPLC (XBridge prep Phenyl 5 μm, 10 × 150 mm; 3.6 mL/min, UV detection at 210 and 280 nm) with a linear H_2_O-CH_3_CN with 0.1% trifluoroacetic acid gradient of 40–70% in 40 min yielding **6** (1.3 mg, t_R_ 32 min) and **7** (3.6 mg, t_R_ 22 min).

The fermentation broth from strain CA-218877 was extracted with EtOAc as mentioned above and MPLC afforded 33 fractions that were combined into five groups according to their LC-UV-MS profiles and those containing the compounds of interest from this chromatography were further purified by preparative and semi-preparative reversed-phase HPLC.

Fraction D (135.4 mg) was chromatographed by semi-preparative reversed-phase HPLC (XBridge prep Phenyl 5 μm, 10 × 150 mm; 3.6 mL/min, UV detection at 210 and 280 nm) with a linear H_2_O-CH_3_CN with 0.1% trifluoroacetic acid gradient of 50–70% in 40 min yielding **3** (5.2 mg, t_R_ 26 min) and **5** (0.6 mg, t_R_ 18 min).

### 4.5. Characterization Data

Phocoenamicin D (**1**): white amorphous solid; ${\left[\alpha \right]}_{D}^{25}$ +1.1 (*c* 0.12, MeOH); UV (DAD) λ_max_ 230, sh 290, 320 nm; IR (ATR) ν_max_ 3376, 2934, 1676, 1445, 1381, 1296, 1203, 1139, 1070, 1025 cm^−1^; for ^1^H and ^13^C NMR data see Table 2; (+)-ESI-TOFMS *m*/*z* 1120. 4884 [M + NH_4_]^+^ (calcd for C_56_H_79_ClNO_20_^+^, 1120.4878).

Phocoenamicin E (**2**): white amorphous solid; ${\left[\alpha \right]}_{D}^{25}$ +6.4 (*c* 0.12, MeOH); UV (DAD) λ_max_ 230, sh 290, 320 nm; IR (ATR) ν_max_ 3388, 2927, 1743, 1677, 1445, 1379, 1293, 1204, 1070, 1024 cm^−1^; for ^1^H and ^13^C NMR data see Table 2; (+)-ESI-TOFMS *m*/*z* 1015.4462 [M + H]^+^ (calcd for C_53_H_72_ClO_17_^+^, 1015.4453).

Maklamicin B (**6**): white amorphous solid; ${\left[\alpha \right]}_{D}^{25}$ −8.5 (*c* 0.12, MeOH); UV (DAD) λ_max_ 230, sh 310 nm IR (ATR) ν_max_ 3384, 2924, 1679, 1617, 1410, 1207, 1137 cm^−1^; for ^1^H and ^13^C NMR data see Table 3; (+)-ESI-TOFMS *m*/*z* 509.3275 [M + H]^+^ (calcd for C_32_H_45_O_5_^+^, 509.3262), 1017.6455 [2M + H]^+^ (calcd for C_64_H_89_O_10_^+^, 1017.6450).

### 4.6. Antibacterial Activity and Cytotoxicity Assay

Compounds **1**–**7** were tested against the growth of Gram-positive bacteria methicillin-resistant *Staphylococcus aureus* (MRSA) MB5393, *Enterococcus faecalis* VANS144492 and *Enterococcus faecium* VANS144754, bacterium *Mycobacterium tuberculosis* H37Ra and Gram-negative bacterium *Neisseria gonorrhoeae* ATCC49226. Moreover, their cytotoxicity against the human liver adenocarcinoma cell line (Hep G2) was evaluated, where the in vitro cell viability was studied based on the MTT (3-(4,5-dimethylthiazol-2-yl)-2,5-diphenyltetrazolium bromide) colorimetric assay [36], using as positive control MMS (Methyl methanesulfonate) at a concentration of 2 mM, causing a 100% cell death. The Hep G2 cell line was obtained from the American Type Culture Collection (ATCC, Manassas, VA, USA). All assays were performed in triplicate, following previously described methodologies [37,38,39].

For the preparation of the samples, each compound was serially diluted in dimethyl sulfoxide (DMSO) with a dilution factor of 2 to provide 10 concentrations starting at 128 μg/mL except for compound **5** which started at 64 μg/mL.

For the antimicrobial assays, the MIC was defined as the lowest concentration of compound that inhibited ≥90% of the growth of a microorganism after overnight incubation, while for the cytotoxicity, the IC_50_ was determined as the concentration that decreases 50% of the cell viability. The Genedata Screener software (Genedata, Inc., Basel, Switzerland) was used to process and analyze the data, as well as calculate the RZ’ factor, which predicted the robustness of the assays [40].

### 4.7. Zebrafish Eleuthero Embryos Toxicity Assay

#### 4.7.1. Zebrafish Care and Maintenance

Adult zebrafish (*Danio rerio*) stocks of AB strain (Zebrafish International Resource Center, Eugene, OR) were maintained as previously described [41]. Briefly, they were kept under a 14/10 h light/dark cycle at 27–28 °C and pH of 6.8–7.5. Fertilized eggs of good quality (fertilized, clear cytoplasm and symmetric cleavage) were selected for the experiments and kept in Petri dishes containing Danieau’s solution (1.5 mM HEPES, 17.4 mM NaCl, 0.21 mM KCl, 0.12 mM MgSO_4_ and 0.18 mM Ca(NO_3_)_2_ and 0.6 μM methylene blue) [42] at 28 °C until compound exposure. All eleuthero embryos were derived from the same spawns of eggs for the comparison between the control and treated groups. Mortality in untreated groups of embryos was <10%. All further experimental work was conducted using Danieau’s solution as the incubation medium.

All procedures were carried out according to the Declaration of Helsinki and conducted following the ARRIVE guidelines [43] and the guidelines of the European Community Council Directive 2010/63/EU, implemented in 2020 by the Commission Implementing Decision (EU) 2020/569 and all the relevant ethical regulations from the Ethics Committee of the University of Leuven (Ethische Commissie van de KU Leuven, approval number ECD 027/2019) and from the Belgian Federal Department of Public Health, Food Safety and Environment (Federale Overheidsdienst Volksgezondheid, Veiligheid van de Voedselketen en Leefmilieu, approval number LA1210261).

#### 4.7.2. Compound Preparation and Toxicity Evaluation

Each of the three compounds (**3**, **4** and **7**) was dissolved in 100% dimethyl sulfoxide (DMSO, spectroscopy grade) and diluted in Danieau’s medium to a final concentration of 25 μΜ (0.5% DMSO), followed by ½ serial dilutions to provide five testing concentrations (1.6, 3.13, 6.25, 12.5 and 25 μM).

Next, 3 dpf zebrafish eleuthero embryos randomly selected were immersed in a 24-well plate containing the three compounds in the different testing concentrations (n = 10 larvae per well). At the same time, 10 Vehicle-treated Control eleuthero embryos (VHC) were also treated with 0.5% DMSO, in accordance with the final solvent concentration of testing compounds to compare the possible effects of the compounds. The eleuthero embryos were incubated under a 14/10 h light/dark cycle at 27–28 °C.

After the 48 h exposure, at 5 dpf, the lethality (cardiac rhythm and degraded body were used as clinical criteria) and touch-stimulation response were assessed using non-anesthetized eleuthero embryos, while the possible morphological defects were evaluated after anesthetizing the eleuthero embryos in 0.5 mM tricaine. All observations were performed using an M80 stereo microscope (Leica Microsystems, Danaher Co., Wetzlar, Germany) and pictures were captured with a Leica DFC310 FX digital color camera (Leica Microsystems, Danaher Co., Germany) and stored.

Mean score of lethality and sub-lethal toxicity per condition was calculated as follows: To each adverse effect observed (impaired motility, bad development, body curvature and swim bladder defects) a score of 1 was given (maximum 4 per larva), and a score of 6 to each dead embryo. The mean score was then calculated for all eleuthero embryos (pooled results) examined per condition, as performed before [41].

#### 4.7.3. Statistical Analysis

The results were obtained from three independent experiments, using ten larvae for each test concentration and compound. Statistical analysis was performed by using GraphPad Prism version 9.0 (GraphPad Software Inc., San Diego, CA, USA) and one-way analysis of variance (ANOVA) followed by Tukey’s multiple comparison test. The criterion for statistical significance was *p* < 0.05.

## 5. Conclusions

Τwo new phocoenamicins (D and E, **1** and **2**, respectively) were isolated from marine strains of *Micromonospora* sp, along with the known phocoenamicin and phocoenamicins B and C (**3**–**5**). Together with the phocoenamicins, maklamicin (**7**) and maklamicin B (**6**) were also obtained, and their production is associated with that of phocoenamicins for the first time. Moreover, the isolation of maklamicin B (**6**) as a natural product is reported for the first time. The two families of compounds share many structural features, as well as common structural variations that are highlighted. One of the new phocoenamicins described (**2**), possessed the characteristic side chain of maklamicin (**7**), unique so far in the phocoenamicin family, indicating a functional group conserved within the two families.

The bioactivity of the eight compounds isolated was evaluated against a panel of human pathogens and they exhibited strong to negligible activities, depending on the compound and the pathogen. As the compounds had small structural differences when compared to each other, the antibacterial assays revealed some structure–activity relationships. Overall, the most active compounds were maklamicin B (**6**) and maklamicin (**7**), which were in some cases more active than the antibiotic used as a positive control, followed by phocoenamicin (**3**) and phocoenamicin B (**4**) against MRSA, *M. tuberculosis* H37Ra, *E. faecalis* and *E. faecium*. The activity against other pathogens should be studied, based on the hypothesis that all-natural products have some receptor-binding function [5].

Finally, the cytotoxicity of all the compounds and the toxicity against zebrafish eleuthero embryos of the three major compounds were evaluated and no broad-spectrum toxicity was detected, making the most active of the compound’s potential candidates for further research and development.

## Figures and Tables

**Figure 1 marinedrugs-21-00443-f001:**
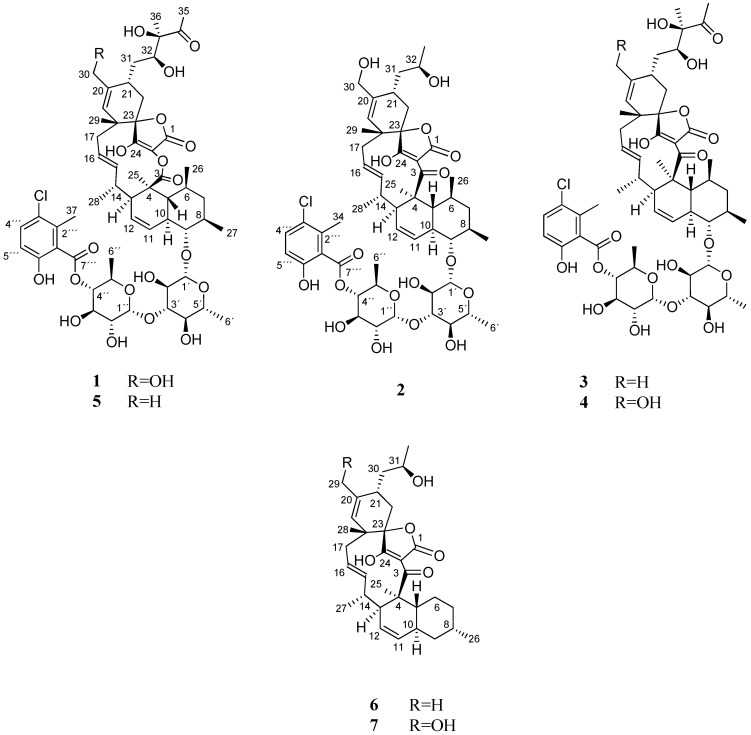
Compounds **1**–**7** isolated from culture broths of *Micromonospora* sp. CA-214671, CA-214658 and CA-218877.

**Figure 2 marinedrugs-21-00443-f002:**
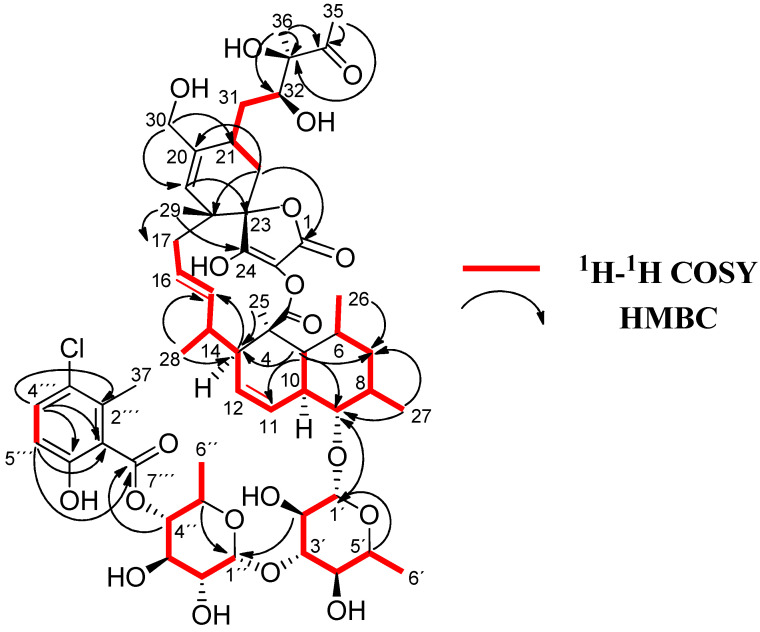
^1^H-^1^H COSY and key HMBC correlations for **1**.

**Figure 3 marinedrugs-21-00443-f003:**
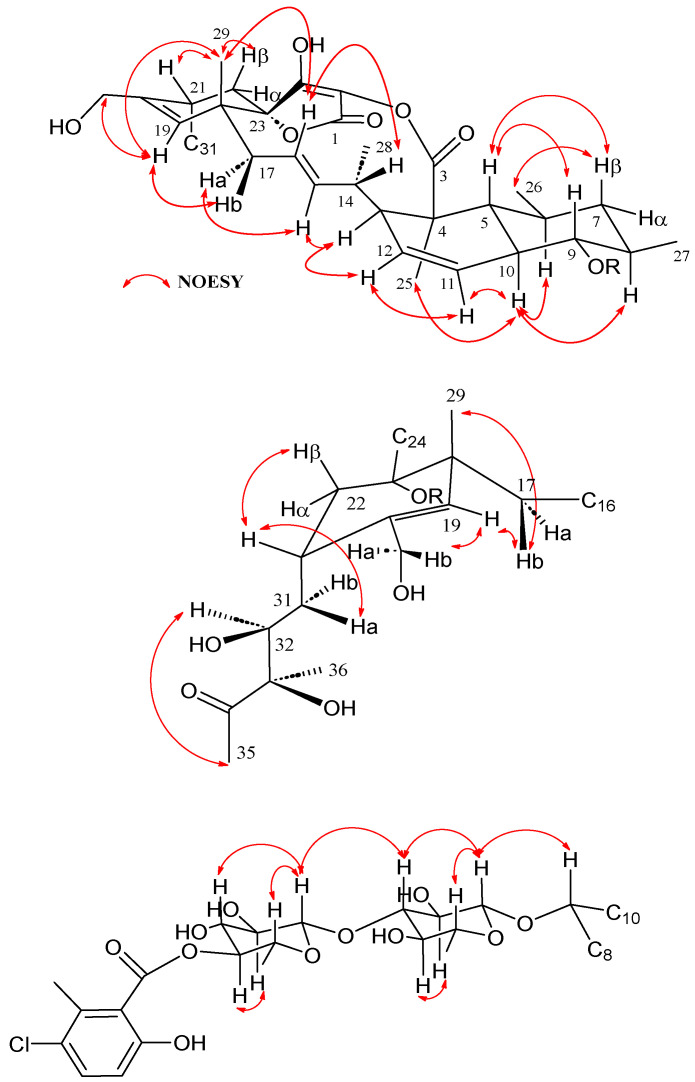
Key NOESY correlations observed in the structure of phocoenamicin D (**1**).

**Figure 4 marinedrugs-21-00443-f004:**
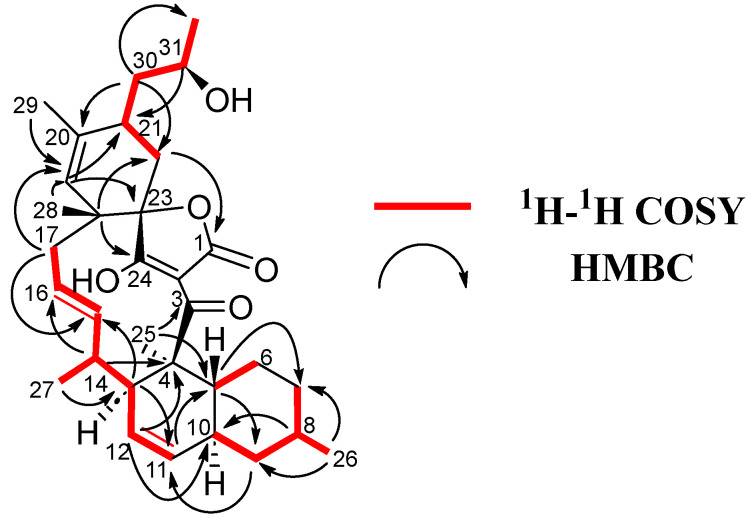
^1^H-^1^H COSY and key HMBC correlations for **6**.

**Figure 5 marinedrugs-21-00443-f005:**
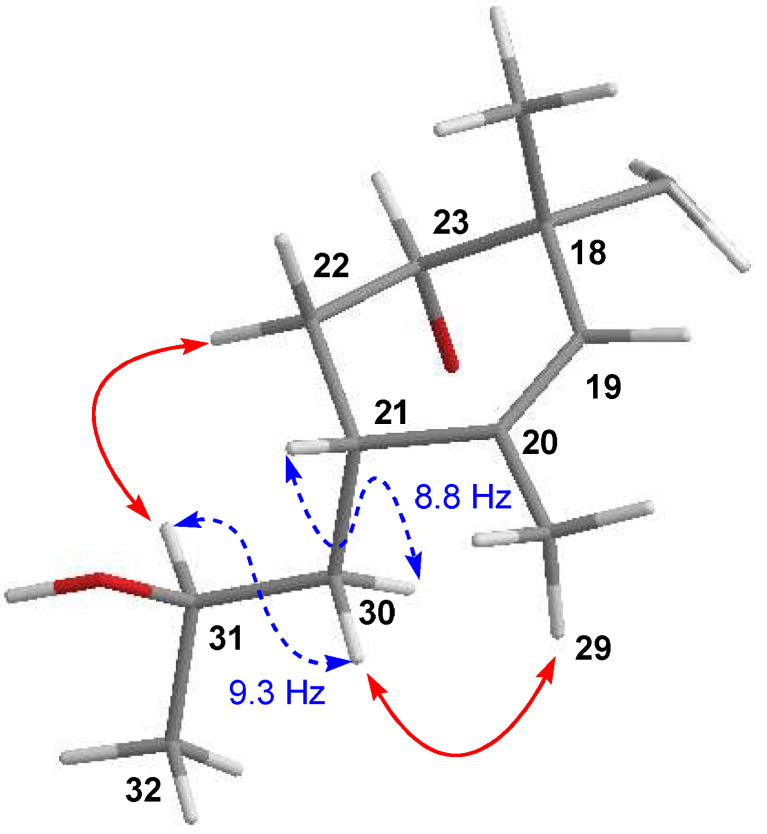
Key coupling constants (blue dashed) and NOESY (red) correlations establishing the configuration at C-31 for **6**.

**Figure 6 marinedrugs-21-00443-f006:**
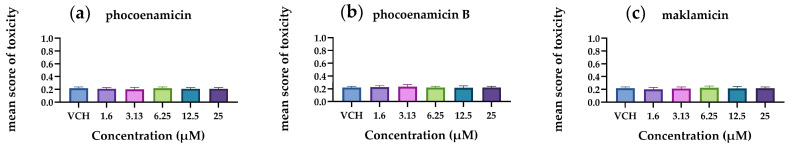
Mean scores of toxicity of the 5 dpf zebrafish eleuthero embryos after their exposure with (**a**) phocoenamicin, (**b**) phocoenamicin B and (**c**) maklamicin in five different concentrations (1.6–25 μM) and the control group (VHC). Three independent experiments were performed, the data were pooled and the mean ± SD was calculated. For the statistical analysis, the mean score of each concentration was compared with the mean scores of every other by using one-way ANOVA with Tukey’s multiple comparison test, *p* < 0.05.

**Figure 7 marinedrugs-21-00443-f007:**
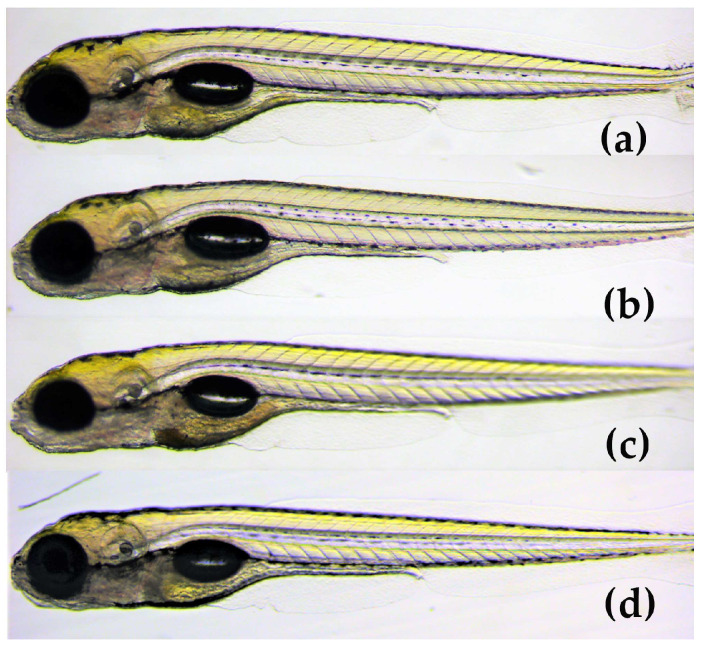
Lateral view of 5 dpf eleuthero-embryos with no developmental abnormalities, after 48 h exposure, (**a**) with 25 μM of phocoenamicin (**b**) with 25 μM of phocoenamicin B, (**c**) with 25 μM of maklamicin and (**d**) Vehicle-treated Control eleuthero embryo (VHC).

**Table 1 marinedrugs-21-00443-t001:** Closest species assignment using the EzBioCloud database, the (%) percentage of similarity, the source of isolation and the geographic origin of each strain.

Strain	Geographic Origin	Ecology	*Micromonospora* Species	Similarity (%)
CA-214658	Gran Canaria, Spain	marine cave sediments	*M. endophytica*	99.63
CA-214671	Gran Canaria, Spain	marine cave sediments	*M. chaiyaphumensis*	99.84
CA-218877	Gran Canaria, Spain	marine invertebrate Porifera	*M. endophytica*	100

**Table 2 marinedrugs-21-00443-t002:** NMR spectroscopic data (500 MHz, CD_3_OD) for phocoenamicins **D** and **E** (**1**–**2**).

	Phocoenamicin D (1)		Phocoenamicin E (2)	
Position	δ_C_, Type	δ_H_, Mult. (*J* in Hz)	δ_C_, Type	δ_H_, Mult. (*J* in Hz)
1	165.6, C	-	186.3, C	-
2	* nd	-	99.7, C	-
3	175.6, C	-	201.5, C	-
4	47.7, C	-	50.5, C	-
5	43.9, CH	1.60, m	43.6, CH	1.81, m
6	38.4, CH	1.53, m	39.2, CH	1.57, m
7α	45.8, CH_2_	1.78, m	45.2, CH_2_	1.79, t (9.6)
7β		1.10, m		1.19, t (9.6)
8	40.7, CH	1.64, m	40.7, CH	1.65 ^a^, s
9	87.9, CH	3.03, t (9.9)	87.8, CH	3.08, t (10.1)
10	48.1, CH	1.94, m	47.9, CH	2.02, m
11	126.8, CH	6.33, t (9.3)	126.9, CH	6.37, d (9.2)
12	127.1, CH	5.65, ddd (9.8, 5.9, 2.4)	125.8, CH	5.60, m
13	51.0, CH	1.99, m	43.9, CH	2.61, m
14	40.7, CH	2.17, m	40.2, CH	2.01, m
15	137.8, CH	4.90, *** w	145.9, CH	5.44, t (11.5)
16	128.5, CH	5.29, dd (14.6, 11.4)	122.2, CH	5.15, dd (14.0, 11.9)
17α	42.2, CH_2_	2.28, m	43.5, CH_2_	2.31, m
17β		1.88, m		1.99, m
18	44.9, C	-	41.0, C	-
19	132.7, CH	5.34, s	131.0, CH	5.34, s
20	139.1, C	-	138.7, C	-
21	30.1, CH	2.68, m	30.7, CH	2.68, m
22α	29.6, CH_2_	1.87, m	30.6, CH_2_	1.85, d (15.4)
22β		2.44, m		2.36 ^b^, s
23	86.5, C	-	88.4, C	-
24	177.5, C	-	206.7, C	-
25	17.4, CH_3_	1.33, ** brs	17.2, CH_3_	1.65 ^a^, s
26	22.6, CH_3_	0.91, ** brs	23.2, CH_3_	0.82, d (6.34)
27	19.9, CH_3_	1.04, d (6.2)	19.9, CH_3_	1.05, d (6.2)
28	22.5, CH_3_	0.91, ** brs	21.5, CH_3_	0.87, d (6.8)
29	23.5, CH_3_	1.33, ** brs	24.1, CH_3_	1.31, s
30α	65.1, CH_2_	4.14, d (13.3)	65.0, CH_2_	4.14, d (13.5)
30β		4.03, d (13.3)		4.03, d (13.5)
31α	33.8, CH_2_	1.76, m	42.4, CH_2_	1.63, m
31β		186, m		1.77, t (9.6)
32	74.0, CH	3.84, dd (11.6, 1.9)	66.0, CH	3.81, m
33	83.4, C	-	24.6, CH_3_	1.18, d (6.1)
34	215.4, C	-	17.8, CH_3_	2.36 ^b^, s
35	25.8, CH_3_	2.24, s	-	-
36	22.0, CH_3_	1.19, s	-	-
1′	104.0, CH	4.35, d (5.4)	103.9, CH	4.36, d (6.8)
2′	75.3, CH	3.46, m	75.2 ^d^, CH	3.46 ^c^, m
3′	88.6, CH	3.48, m	88.5, CH	3.46 ^c^, m
4′	75.6, CH	3.11, t (8.8)	75.5, CH	3.12, t (8.6)
5′	72.9, CH	3.25, m	72.7, CH	3.24, m
6′	18.3, CH_3_	1.28, d (6.2)	18.3, CH_3_	1.28, d (5.8)
1″	105.4, CH	4.61, d (7.7)	105.2, CH	4.61, d (7.8)
2″	76.0, CH	3.43, t (8.3)	75.9, CH	3.43, t (7.8)
3″	75.3, CH	3.65, t (9.4)	75.2 ^d^, CH	3.64, t (9.2)
4″	77.8, CH	4.88, *** w	77.8, CH	4.90, *** w
5″	71.7, CH	3.69, m	71.6, CH	3.69, m
6″	18.0, CH_3_	1.35, d (6.1)	18.0, CH_3_	1.36, d (5.3)
1‴	124.4, C	-	124.3, C	-
2‴	135.6, C	-	135.4, C	-
3‴	126.0, C	-	125.9, C	-
4‴	132.4, CH	7.25, d (8.8)	132.3, CH	7.26, d (8.3)
5‴	115.9, CH	6.71, d (8.8)	115.9, CH	6.71, d (7.9)
6‴	155.3, C	-	155.1, C	-
7‴	169.2, C	-	169.3, C	-
37	17.9, CH_3_	2.36, s	-	-

* nd = not detected; ** brs = broad signal; *** w = obscured by the water peak; ^a,b,c,d^ overlapping signals.

**Table 3 marinedrugs-21-00443-t003:** NMR spectroscopic data (500 MHz, CD_3_OD) for maklamicin B (**6**) and maklamicin (**7**).

	Maklamicin B (6)		Maklamicin (7)	
Position	δ_C_, Type	δ_H_, Mult. (*J* in Hz)	δ_C_, Type	δ_H_, Mult. (*J* in Hz)
1	* nd	-	169.5, C	-
2	100.2, C	-	100.2, C	-
3	202.4, C	-	202.5, C	-
4	52.1, C	-	52.1, C	-
5	43.9, CH	1.50, dd (8.3, 12.5)	43.9, CH	1.51, dd (8.3, 10.3)
6α	24.4, CH_2_	1.24, m	24.4, CH_2_	1.29, m
6β		2.08, m		2.08, m
7α	34.1, CH_2_	1.57, m	34.0, CH_2_	1.60, m
7β		1.75, m		1.77, m
8	29.2, CH	2.07, m	29.3, CH	2.07, m
9α	40.9, CH_2_	1.62, m	40.8, CH_2_	1.64, m
9β		1.42, d (4.8)		1.43, d (4.9)
10	34.3, CH	2.04, m	34.3, CH	2.05, m
11	131.3, CH	5.41, d (9.3)	131.9, CH	5.40, d (10.0)
12	126.3, CH	5.48, ddd (9.9, 6.2, 2.4)	126.2, CH	5.49, ddd (10.0, 6.3, 2.3)
13	42.5, CH	2.85, m	42.6, CH	2.84, m
14	41.2, CH	1.88, m	41.3, CH	1.89, m
15	144.9, CH	5.43, m	145.2, CH	5.46, m
16	122.9, CH	5.07, ddd (13.6, 11.8, 2.5)	122.7, CH	5.09, ddd (13.9, 11.6, 2.7)
17α	43.7, CH_2_	2.30, dd (14.3, 10.9)	43.4, CH_2_	2.33, dd (14.7, 8.0)
17β		1.95, d (14.5)		2.02, d (14.5)
18	40.6, C	-	40.5, C	-
19	130.9, CH	5.03, s	131.1, CH	5.33, s
20	135.0, C	-	138.9, C	-
21	34.5, CH	2.42, br dd (8.8, 7.2)	30.5, CH	2.67, br dd (7.7, 7.4)
22α	30.9, CH_2_	1.77, m	31.0, CH_2_	1.83, d (15.6)
22β		2.33, dd (14.3, 7.2)		2.33, dd (14.7, 8.0)
23	87.9, C	-	88.2, C	-
24	205.0, C	-	204.9, C	-
25	16.3, CH_3_	1.56, s	16.4, CH_3_	1.58, s
26	19.5, CH_3_	1.06, d (7.2)	19.4, CH_3_	1.07, d (7.7)
27	21.3, CH_3_	0.87, d (7.1)	21.2, CH_3_	0.88, d (7.5)
28	24.4, CH_3_	1.24, s	24.1, CH_3_	1.29, s
29α	22.3, CH_3_	1.75, s	65.1, CH_2_	4.16, d (13.5)
29β				4.01, d (13.5)
30α	42.6, CH_2_	1.63, dd (14.0, 11.1)	42.7, CH_2_	1.62, m 1.79, m
30β		1.75, m		
31	65.9, CH	3.78, dq (9.3, 6.6)	66.1, CH	3.79, dq (9.6, 6.7)
32	24.6, CH_3_	1.17, d (6.2)	24.6, CH_3_	1.17, d (6.3)

* nd = not detected.

**Table 4 marinedrugs-21-00443-t004:** Antimicrobial bioassay results (MIC, μM) against MRSA, *M. tuberculosis* H37Ra, *E. faecium*, *E. faecalis* and *N. gonorrhoeae* and cytotoxic activities (IC_50_, μM) against the Hep G2 cell line of the 7 spirotetronates isolated.

		MIC (μM)							
Compounds		(1)	(2)	(3)	(4)	(5)	(6)	(7)	Control *
Pathogen	Strain								
MRSA	MB5393	58	31.5–63	3.7	7.4	29.4	2	<0.5	1.4 (V)
*M. tuberculosis*	H37Ra	58	31.5–63	7.5	14.7	29.4–58.8	2	61	2.8–5.4 (S)
*E. faecalis*	VANS 144492	>58	>63	7.5	>58.8	>58.8	62.9	1.9	1.4 (V)
*E. faecium*	VANS 144754	≥116	>63	3.7–7.5	≥58.8	>58.8	1	1	1.4 (V)
*N. gonorrhoeae*	ATCC49226	>116	>126	>119.4	>117.7	>58.8	>251.6	>244	5.6 (P)
		**IC_50_ (μM)**							
Hep G2 (liver) cell line		-	21	-	17.6	-	-	40.2	

* (V) vancomycin, (S) streptomycin, (P) penicillin.

## Data Availability

Additional data are available in the Appendix A.

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
