# Peer review of "New Phocoenamicin and Maklamicin Analogues from Cultures of Three Marine-Derived Micromonospora Strains"

_marinedrugs, 2023, doi:10.3390/md21080443_

Round 1
Reviewer 1 Report
Manuscript (marinedrugs-2519368) reports on the chemical study of three Micromonospora strains identified as producers of phocoenamicins. The strains CA-214671, CA-214658 and CA-218877 were scaled up and the EtOAc extracts afforded the new phocoenamicins , pho-coenamicins D (1) and E (2) , along with the known phocoenamicin, phocoenamicins B and C, as well as maklamicin and maklamicin C(3-7). The isolation and structural elucidation of compounds 1,2 and 6 (isolated as a natural product for the first time) are reported. Biological activity was evaluated. All the isolated compounds were tested against five human pathogens. Cell viability evaluated against the human liver adenocarcinoma cell line (HepG2). Additionally, an in vivo test of against zebrafish eleuthero embryos of compounds phocoenamicin, phocoenamicin B and maklamicin demonstrated no toxicity.
In general terms, the manuscript is correctly written and only a few aspects should be considered:
- Structure of compound 1 in figures 1 and 2 must be revised. The O of the carbonyl ester of the macrocyclic lactone should be attached to C-2 of the tetronic acid instead of being linked to C-4.
- Figures 2 and 3 seem extremely crowded with HMBC correlations and, in consequence, it lacks clarity. It is suggested to simplify just with key correlations.
- Section 2.4 on antimicrobial and cytotoxic activity. For pure compounds, it is recommended to report MIC and IC50 values in uM concentration instead of ug/mL, which is used for extracts and fractions. In this regard, the last paragraph of section 2.5 should be revised accordingly.
- Regarding the structure-activity relationship, no comments have been mentioned to the glicosidic side chain of compounds, which seem to be a key structural feature for the biological activity. For instance, lack of that fragment in maklamicins 6 and 7 seem to increase the antimicrobial activity compared to that of phocoenamicins in some of the tested pathogens.
- Discussion section, paragraph 2, last sentence should be revised. It seems not complete.
- Section 4.3, line 9: “The content of the three flasks…”
- Complete reference 13 (Journal and DOI) and 14, 26, 28 (DOI)
Author Response
In general terms, the manuscript is correctly written and only a few aspects should be considered:
- Structure of compound 1 in figures 1 and 2 must be revised. The O of the carbonyl ester of the macrocyclic lactone should be attached to C-2 of the tetronic acid instead of being linked to C-4.
The structure of the compound has been corrected, thank you for noting it out.
- Figures 2 and 3 seem extremely crowded with HMBC correlations and, in consequence, it lacks clarity. It is suggested to simplify just with key correlations.
The HMBC correlations appearing in Figures 2 and 3 were simplified to key correlations.
- Section 2.4 on antimicrobial and cytotoxic activity. For pure compounds, it is recommended to report MIC and IC50 values in uM concentration instead of ug/mL, which is used for extracts and fractions. In this regard, the last paragraph of section 2.5 should be revised accordingly.
The MIC and IC50 concentration values have been changed to μM where it appeared throughout the article. The last paragraph of section ".5 has been modified.
- Regarding the structure-activity relationship, no comments have been mentioned to the glicosidic side chain of compounds, which seem to be a key structural feature for the biological activity. For instance, lack of that fragment in maklamicins 6 and 7 seem to increase the antimicrobial activity compared to that of phocoenamicins in some of the tested pathogens.
A comment was added in Discussion, page 13.
‘’Overall, the lack of the glycosidic side chain in maklamicins (6) and (7) seems to increase the antimicrobial activity compared to that of phocoenamicins in some of the tested pathogens.‘’
- Discussion section, paragraph 2, last sentence should be revised. It seems not complete.
It was corrected.
- Section 4.3, line 9: “The content of the three flasks…”
It was corrected, thank you for the observation.
- Complete reference 13 (Journal and DOI) and 14, 26, 28 (DOI)
References 13, 14, 26 and 28 have been corrected.
Reviewer 2 Report
Dear Authors,
Good chemical and pharmaceutical research work developed. Congratulations
Author Response
We that the reviewer for the time spent to review our article and acknowledge his/her positive comments
Reviewer 3 Report
This manuscript describes the isolation of two new phocoenamicins, phocoenamicins D and E, along with five known compounds from three marine-derived micromonospora Strains. The structures of the new compounds were elucidated by extensive spectroscopic methods. The antibacterial activity, cytotoxicity and zebrafish eleuthero embryos toxicity of isolated compounds was also evaluated. The experimental data and spectra of compounds were clear. Before the current text could be accepted, there are a number of, but not limited to, issues that the authors should address:
1. Abstract: which compound was reported for the first time as a natural product?
2. The structure of compounds 3-5 should be presented in Figure 1.
3. Page 6: second paragraph, line 3, “C-5′” should be “C-3′”, and the absolute configuration of compound 1 can’t just be proposed.
4. Page 6: Table 3, H-5 of compound 7 lacks peak pattern.
5. Page 8, the last paragraph, the HMBC correlations from H-25 to C-7 seems impossible, please double check it.
6. Page 9: third paragraph, since the signals were overlapped, the relative configuration of 6 can’t just be tentatively assigned.
7. What’s the positive control for cytotoxicity assay?
8. Page 9: the first paragraph of Disscussion, 29-deoxymaklamicin (6) should be consistent as maklamicin B, same issue in Conclusions. “29-deoxymaklamicin (7) was previously reported as” should be revised as “maklamicin B (6) was ….” .
9. References: Please list them in alphabetical order.
10. Supplementary information: The 1H NMR spectra lacked integration and peak positions. The 13C NMR spectra lacked peak positions. And the 2D NMR were too small.
It's fine。
Author Response
- Abstract: which compound was reported for the first time as a natural product?
A phrase was added to make the sentence more clear
´´…as well as maklamicin and maklamicin B, the latter being reported for the first time as a natural product.´´
- The structure of compound s3-5 should be presented in Figure 1.
The structure of compounds 3-5 was added in Figure 1.
- Page 6: second paragraph, line 3, “C-5′” should be “C-3′”, and the absolute configuration of compound 1 can’t just be proposed.
It is indeed C-3′ and it was corrected, thank you for noting it out.
- Page 6: Table 3, H-5 of compound 7 lacks peak pattern.
The multiplicity of the signal has been added, thank you for the observation.
- Page 8, the last paragraph, the HMBC correlations from H-25 to C-7 seems impossible, please double check it.
This is a mistake, The HMBC correlation is not from H-25 but from H-5 to C-7. It has been corrected
- Page 9: third paragraph, since the signals were overlapped, the relative configuration of 6 can’t just be tentatively assigned.
It was rephrased stating that the relative configuration for the tetronic acid chiral center of both molecules was tentatively assigned as S*, in agreement with that of the previously reported spirotetronate antibiotics.
- What’s the positive control for cytotoxicity assay?
The positive control was MMS (Methyl methanesulfonate) and was added in the methods paragraph 4.6, page 16.
- Page 9: the first paragraphof Disscussion, 29-deoxymaklamicin (6) should be consistent as maklamicin B, same issue in Conclusions. “29-deoxymaklamicin (7) was previously reported as” should be revised as “maklamicin B (6) was ….” .
It was changed, thank you for the observation.
- References: Please list them in alphabetical order.
References are listed according to the Marine Drugs guidelines
‘’References: References must be numbered in order of appearance in the text (including table captions and figure legends) and listed individually at the end of the manuscript.’’
https://www.mdpi.com/journal/marinedrugs/instructions
- Supplementary information: The 1H NMR spectra lacked integration and peak positions. The 13C NMR spectra lacked peak positions. And the 2D NMR were too small.
It has been changed.